# Micafungin-Induced Cell Wall Damage Stimulates Morphological Changes Consistent with Microcycle Conidiation in *Aspergillus nidulans*

**DOI:** 10.3390/jof7070525

**Published:** 2021-06-29

**Authors:** Samantha Reese, Cynthia Chelius, Wayne Riekhof, Mark R. Marten, Steven D. Harris

**Affiliations:** 1School of Biological Sciences, University of Nebraska, Lincoln, NE 68503, USA; samantha.reese@huskers.unl.edu (S.R.); wriekhof2@unl.edu (W.R.); 2Biologics Development, Global Product Development and Supply, Bristol Myers Squibb Company, Devens, MA 01434, USA; Cindy.chelius@bms.com; 3Department of Chemical, Biochemical and Environmental Engineering, University of Maryland Baltimore County, Baltimore, MD 21250, USA; marten@umbc.edu; 4Department of Plant Pathology and Microbiology and Department of Entomology, Iowa State University, Ames, IA 50011, USA

**Keywords:** *Aspergillus nidulans*, micafungin, cell wall integrity signaling pathway, BrlA, RNA-Sequencing, microcycle conidiation

## Abstract

Fungal cell wall receptors relay messages about the state of the cell wall to the nucleus through the Cell Wall Integrity Signaling (CWIS) pathway. The ultimate role of the CWIS pathway is to coordinate repair of cell wall damage and to restore normal hyphal growth. Echinocandins such as micafungin represent a class of antifungals that trigger cell wall damage by affecting synthesis of β-glucans. To obtain a better understanding of the dynamics of the CWIS response and its multiple effects, we have coupled dynamic transcriptome analysis with morphological studies of *Aspergillus nidulans* hyphae in responds to micafungin. Our results reveal that expression of the master regulator of asexual development, BrlA, is induced by micafungin exposure. Further study showed that micafungin elicits morphological changes consistent with microcycle conidiation and that this effect is abolished in the absence of MpkA. Our results suggest that microcycle conidiation may be a general response to cell wall perturbation which in some cases would enable fungi to tolerate or survive otherwise lethal damage.

## 1. Introduction

The cell wall is an important feature of filamentous fungi, where it is responsible for the protection of hyphal integrity and the maintenance of hyphal morphology [1]. In fungi, damage to the cell wall poses a serious threat to cell viability. To mitigate this possibility, fungi employ cell surface sensors that detect damage to the cell wall and activate a suite of mechanisms that repair the damage and enable the resumption of normal growth [1]. This response is known as the Cell Wall Integrity Signaling (CWIS) pathway [2]. This pathway is found in many fungal species and key signaling features includes the Rho1 GTPase, protein kinase C, and the CWIS MAP kinase cascade [2]. In Aspergillus nidulans, the pathway is conserved and terminates with the MAP kinase MpkA [3].

There are many perturbations that can trigger the CWIS pathway in fungi including antifungals detected in the environment. Micafungin, an echinocandin antifungal drug, has been shown to activate the CWIS pathway. It acts by inhibiting the activity of β-1,3 glucan synthase, thereby depleting β-1,3 glucan from the cell wall and compromising hyphal integrity [4]. This leads to hyphal tip bursting and a decrease in internal hydrostatic pressure of the hyphae [5]. The CWIS response to micafungin is well understood in yeast but is less studied in filamentous fungi such as *A. nidulans*. Notably, the transcriptional output of the CWIS pathway differs significantly in *A. nidulans* than in the yeast Saccharomyces cerevisiae [3]. To better understand the dynamic nature of the CWIS in *A. nidulans*, we have employed a “multi-‘omics” approach aimed at characterizing key effectors and outputs of micafungin perturbation [6].

The asexual life cycle of filamentous fungi is important for maintaining species viability in the face of adverse environmental conditions [7]. The formation of spores via asexual means is typically a well-regulated global response to external stimuli [8]. However, under some circumstances, such as specific environmental triggers, the normal regulatory and developmental processes that underlie asexual sporulation can be short-circuited to allow rapid production of spores [9]. The shortened cycle for the production of conidia by newly germinated spores without any further hyphal growth is termed microcycle conidiation [10]. At this time, the overall benefits of this process and the extent to which fungi use it as a short-term survival strategy remain unclear. Microcycle conidiation has been observed across many ascomycetes, including members of the genera Aspergillus, Fusarium, and Metarhizium [8,9,11]. In these fungi, the genetic pathways that regulate microcycle conidiation and coordinate its occurrence relative to environmental conditions are poorly understood in limited detail.

The objective of this study was to gain a deeper understanding of the filamentous fungal response to cell wall damage by exposing the model fungus A. nidulans response to micafungin exposure as a paradigm. Upon exposure of growing hyphae to micafungin, samples were taken over a time course and subjected to both phosphoproteomic and transcriptomic analyses [6]. Here, we describe the impacts of exposure to micafungin on patterns of gene expression in both wildtype and ∆mpkA mutants. Our results suggest that the response to cell wall damage is attenuated in ∆mpkA mutants and reveal specific classes of genes whose expression is dependent upon MpkA. Notably, this includes the master regulator of conidiation, brlA. Indeed, we show that morphological changes consistent with microcycle conidiation are triggered by exposure to micafungin, thereby raising the possibility that microcycle conidiation might be a critical feature of the response to anti-fungal drugs.

## 2. Materials and Methods

### 2.1. Media and Growth Conditions

All strains used in this study are described in Appendix A. Standard media and growth conditions were employed [12,13]. Media included Yeast Extract-Glucose-Vitamin (YGV; 0.5% yeast extract, 1% glucose, 0.1% vitamins) [13], and Malt Extract-Glucose (MAG; 2% malt extract, 2% glucose, 0.2% peptone, 0.1% trace elements, 0.1% vitamins) [13].

### 2.2. Global Analysis Strain Generation, Growth Conditions, RNA Extraction, and RNA Sequencing

The generation of all strains, growth conditions for RNA-Sequencing samples, RNA extraction, and RNA Sequencing pipeline can be found in our previous paper [6]. Note that the RNA-Sequencing bioinformatics pipeline, DESeq2 1.20.1, was the bioinformatics package used to analyze the read counts generated by HISAT2 2.1.0 and HTSeq-Counts 0.9.1. The FDR is calculated for every gene though DESeq2 at every time point and is called “adjusted *p*-value”. The adjusted *p*-value is what was used in addition of the Log2Fold (+/−) 2 cut off to generated the DE genes list used in this study’s investigation. Significance for the RNA-Sequencing gene expression was also based on our previous paper with a less strenuous adjusted *p*-value of ≤0.1. This was done to facilitate the incorporation of proteomic and transcriptomic data into a predictive gene expression model. NCBI Sequence Read Archive Accessions can be found in (Appendix A).

### 2.3. QRT-PCR for RNA-Sequencing Verification

Wildtype and ΔmpkA hyphae grown in YGV for 11 h were exposed to 1 ng of micafungin per 1 mL growth media for 75 min [12]. Control hyphae were left untreated [12]. The hyphae were frozen with liquid nitrogen, and RNA was then extracted and purified. A cDNA library was generated using the cDNA Library Kit from Thermo Fisher Scientific (catalog A48571) and the Reverse transcription PCR kit from Millipore Sigma, St. Louis, Mo USA (Product No. HSRT100), with specific primers designed for each target gene (Appendix A). The samples were analyzed using Bio-Rad CFX Manager 3.1. There were three biological replicates, and the fold-change was determined by comparing the treated and untreated samples using the ∆∆Ct method [14]. Histone H2B was used as a control for qRT-PCR normalization.

### 2.4. Microscopy

Images were collected using either an Olympus BX51 microscope with a reflected fluorescence system fitted with a Photometrics CoolSnap HQ camera coupled to Lambda B10 Smart Shutter control system (Sutton Instruments), or an EVOS FL microscope (Advanced Microscopy Group). Images were initially processed using MetaMorph software (Molecular Devices).

### 2.5. Micafungin Concentration Experiments

Wildtype and ΔmpkA hyphae were inoculated and grown as mentioned above on cover slips in 60 mm × 15 mm Petri dishes for 11 h in 25 mL of YGV media. At this time, micafungin (0.0 ng/mL, 0.01 ng/mL, 0.1 ng/mL, 1.0 ng/mL) was added and hyphae grown for an additional 3 h before viewing.

### 2.6. Microcycle Conidiation Experiment with Wildtype and ΔmpkA

Wildtype and ΔmpkA hyphae were inoculated and grown as mentioned above with the micafungin dosage of 0.1 ng/mL in a total of 25 mL of YGV media. Hyphae were grown an additional 9 h post antifungal addition.

### 2.7. Up-Regulation of brlA Experiments

The alcA::brlA strain constructed by Adams et al. (1988) was grown on cover slips in YGV for 11 h, then the cover slips were shifted into MNV 100 mM L-Threonine to induce the alcA promoter [15,16]. The strains were allowed to grow an additional 3 h in MNV 100 mM L-Threonine media then imaged [12].

## 3. Results

### 3.1. Global Transcript Expression

To better understand the dynamic response of the CWIS pathway to cell wall damage, hyphae exposed to micafungin were harvested at 0, 5, 10, 15, 20, 25, 30, 40, 50, 60, 75, 90, 120 min, and subjected to transcriptomic analysis as described in Chelius et al. (2020) [6]. The zero-minute time point had no micafungin exposure and served as our untreated control for transcript expression. Significantly expressed genes (*p*-value ≤ 0.1) were identified following analysis using DESeq2 in RStudio [17]. The wildtype strain showed an exponential increase in significantly expressed genes beginning at 25 min through to the last time point with the total of 1614 genes showing altered expression (Figure 1A and Appendix A). The ΔmpkA strain showed a linear increase in significantly expressed genes from beginning at 30 min through to the last time point with a total of 964 significantly expressed genes (Figure 1A and Appendix A).

There was a linear increase in both up and down-regulated Differently Expressed (DE) genes (adjusted *p*-value ≤ 0.1 and log2fold (+/−) 2.0) in both wildtype and ΔmpkA starting at the 30 min time point (Figure 1B,C and Appendix A). There also were more up-regulated DE genes in both wildtype and ΔmpkA strains. In wildtype, 274 DE genes were up-regulated and 55 DE genes down-regulated, whereas in ΔmpkA 264 DE genes were up-regulated and 24 DE down-regulated (Figure 1B,C and Appendix A). Some of the DE genes were up-regulated in both wildtype and ΔmpkA (Appendix A). The first shared DE genes were AN1199 (Uncharacterized), AN3888 (Uncharacterized), and AN8342 (Uncharacterized) at time point 50 min. The number of shared DE genes increased throughout the time course, such that at the last time point there were eighty-five shared genes between wildtype and ΔmpkA. These genes seemingly define a large and previously unrecognized component of the CWIS that is independent of MpkA.

Analysis of DE genes in both wildtype and the ΔmpkA strain at each time point revealed that a number of apparent gene clusters (solely defined as three or more genes with consecutive ANID numbers) were differentially regulated upon exposure to micafungin (Appendix A). For example, as early as 40 min post micafungin exposure, expression of the AN5269-5273 cluster was up-regulated in an MpkA-dependent manner whereas that of the AN7952-7955 cluster was up-regulated even in the absence of MpkA. In total, ten of the 19 presumptive gene clusters could be sorted into three groups whose expression was affected by micafungin; (i) MpkA-dependent micafungin induced (three), (ii) MpkA-independent micafungin-induced (four), and (iii) MpkA-dependent micafungin-repressed (three). Moreover, seven of these ten DE clusters responded within the first 60–75 min post-exposure. The remaining nine DE clusters were dependent on MpkA for expression but were not affected by exposure to micafungin and were similar to those identified in our earlier study of MpkA-dependent gene expression in the absence of cell wall stress [18]. With some exceptions (Andersen et al., 2013), the precise function of the apparent micafungin-induced gene clusters is unknown, but this observation highlights the complexity of the CWIS and the presence of both MpkA-dependent and -independent components [10].

### 3.2. GO Term Analysis

The distribution of DE genes amongst Gene Ontology (GO) categories were analyzed using the Aspergillus Database [19,20]. In wildtype the up-regulated GO term categories showed a shift in gene expression starting at the 60 min time point (Figure 2A and Appendix A). From the 25 min time point to the 60 min time point, the data revealed a response to micafungin that featured increases in oxidoreductase activity genes including; AN7708 (predicted NADP+ 1-oxidoreductase activity) (Log2Fold 2.25), AN3043 (predicted role in oxidation-reduction process) (Log2Fold 2.66), AN9375 (predicted oxidoreductase activity and role in oxidation-reduction process) (Log2Fold 2.71), uaZ (urate oxidase) (Log2Fold 2.19), and aoxA1 (urate oxidase) (Log2Fold 2.90) (Figure 2A and Appendix A). However, at the 75 min time point, additional GO term categories such as light-induced genes (brlA µORF (brlA regulator leader sequence), brlA (zinc finger transcription factor), prtA (thermostable alkaline protease), and AN0045 (uncharacterized)) are up-regulated. The last time point had nine light-induced DE up-regulated genes (brlA, prtA, AN0045, AN0693 (uncharacterized), silA (uncharacterized), AN3304 (GABA transporter), AN3872 (uncharacterized), conJ (conidiation gene), AN8641(uncharacterized) (Figure 2A and Appendix A). Additionally, there is a known connection with secondary metabolism and light induced genes, which is particularly evident at the 75 min time point and beyond. Examples mdpE (zinc finger transcription) (Log2Fold 2.55), pkdA (polyketide synthase) (Log2Fold 4.31), AN6962 (predicted secondary metabolism gene cluster member) (Log2Fold 4.38), AN9314 (protein with homology to entkaurene synthases) (Log2Fold 5.79), and AN2116 (predicted catalytic activity) (Log2Fold 6.34) (Figure 2A and Appendix A) [21]. Notably, the distribution of DE up-regulated GO terms for the ΔmpkA mutant does not display this apparent shift, as there are no light-induced up-regulated in any of the time points of the ΔmpkA strain (Figure 2A and Appendix A).

### 3.3. Light Induced Gene Induction

Further investigation of genes that exhibit increased expression upon exposure to micafungin in wildtype revealed the induction of brlA (AN0973) and brlA μORF (AN0974) at 75, 90, and 120 min (Appendix A). brlA μORF is a small ORF located upstream ORF of brlA in a leader sequence. The μORF regulates conidial development by repressing the beta transcript of brlA when external cues trigger the initiation of asexual development [22]. brlA was up-regulated with Log2Fold change of 2.70, 3.11, 3.50 at time points 75, 90, 120 min, respectively (Figure 3A and Appendix A). brlA μORF was up-regulated with Log2Fold changes of 3.5, 3.2, 3.1 at time points 75, 90, 120 min, respectively (Figure 3A and Appendix A). qRT-PCR results at 75 min after micafungin exposure confirmed the significant up-regulation of brlA (7.8 log fold change) and brlA μORF (12.09 log fold change) in the presence of micafungin in wildtype but not in the mpkA knockout (brlA 0.91 log fold change; brlA μORF 1.52 log fold change) (Figure 3B).

The induction of brlA expression in wildtype hyphae exposed to cell wall damage was unexpected. brlA encodes a C2H2 zinc finger transcription factor that regulates conidiophore development [15]. However, asexual structures typically do not develop in aerated shaking flask cultures as were used in this study (i.e., hyphae should only grow in a vegetative state) [23]. In the well-characterized regulatory pathway that controls asexual development in A. nidulans, upstream regulators activate brlA expression, which then results in the induced expression of the downstream transcription factors abaA and wetA [15]. Additionally, there was no significant increase in transcript expression for abaA (AN0422) (Log2Fold 0.329 at time point 75 min) or wetA (AN1937) (Log2Fold 0.290 at time point 75 min) upon exposure to micafungin (Appendix A), nor were any know upstream activators affected (data not shown). These results suggest that the observed increased expression of brlA does not reflect the full induction of the canonical asexual development regulatory pathway.

### 3.4. Hyphal Response to Micafungin Exposure

To investigate the possibility that expression of brlA might correlate with specific morphological changes. We exposed wildtype and ΔmpkA hyphae to 0 ng/mL (untreated control), 0.01 ng/mL, 0.1 ng/mL, and 1.0 ng/mL of micafungin in YGV media after they had already been growing for 11 h on cover slips at 28 °C. We let the strains grow under these same conditions for an additional 3 h post-treatment. Whereas the untreated control strains exhibited no morphological alterations (Figure 4), wildtype hyphae exposed to micafungin possessed bulges at hyphal tips and lateral branching points regardless of micafungin dosage (Figure 4).

The bulging phenotype of wildtype hyphae exposed to micafungin resembled a pattern of attenuated spore development in filamentous fungi that is known as microcycle conidiation [9]. This phenomenon has been observed in Aspergilli exposed to poor growth substrates such as indoor construction materials or hospital air filters [24]. To test the idea that microcycle conidiation might be a response to micafungin exposure, we repeated the previous experiment but extended the post-dosage incubation period to allow for spore formation. Accordingly, wildtype and ΔmpkA hyphae were grown and treated with micafungin as described above, with the difference that hyphae were incubated for 9 h instead of three prior to observation. In untreated control hyphae, no morphological alterations were observed (Figure 5). However, wildtype hyphae exposed to 0.1ng/mL of micafungin appeared to exhibit development of spores at hyphal tips (Figure 5). The formation of these spores was relatively infrequent, with generally less than 5% of the hyphal tips displaying terminal bulges. Unlike wildtype hyphae, the ΔmpkA strain did not exhibit development of spore-like bulges following 9 h of micafungin exposure (Figure 5).

### 3.5. Similar Spore Morphologies upon Induced brlA Expression and Post Micafungin Exposure

We sought to determine whether the spores generated by micafungin-induced microcycle conidiation are similar to those that result from established conditions known to trigger this process. Previously, it was demonstrated that forced activation of brlA expression results in production of a single spore at hyphal tips and branching sites [15,16]. As this is effectively a form of microcycle conidiation, we compared spores produced upon forced induction of brlA to those generated as a result of micafungin exposure. The alcA::brlA strain was grown on cover slips for 3 h in YGV media at 280 °C then shifted to MV 100 mM L-Threonine to grow for an additional 3 h to induce brlA expression. The wildtype strain was grown and exposed to micafungin for nine hours by the same methods as described above. Both strains were viewed under the microscope. The alcA::brlA strain developed spores at the hyphal tips, as did the wildtype strain exposed to micafungin (Figure 6). This suggest that BrlA-induced microcycle conidiation might be a key response to cell wall damage caused by micafungin in *A. nidulans* hyphae.

## 4. Discussion

The purpose of this study was to investigate the dynamic transcriptional response of A. nidulans to the antifungal drug micafungin, which compromises cell wall integrity through the depletion of β-1,3-glucans. We compared the response of wildtype hyphae to that of an ΔmpkA deletion mutant, in which the terminal kinase of the CWIS pathway has been disabled. In a prior study, our characterization of the dynamic phosphoproteomic and transcriptional response of wildtype hyphae revealed candidate signaling and morphogenetic pathways that promote repair of damaged cell walls and the maintenance of hyphal integrity [6]. Here, we show that a significant transcriptional response still occurs in the absence of MpkA. Nevertheless, MpkA is required for expression of a set of genes that seemingly support the long-term survival of A. nidulans in the face of cell wall damage. Unexpectedly, amongst these genes is the master regulator of asexual development, brlA. Further investigation suggested that expression of BrlA elicits morphological changes consistent with microcycle conidiation, which might represent a heretofore unconsidered strategy used by fungi to resist the impacts of cell wall damage.

The distinct features of the A. nidulans CWIS pathway compared to the established yeast paradigm were first noted by Fujioka et al. (2007), who observed that the response in filamentous fungi includes a significant MpkA-independent component [3]. They found that expression of several cell wall-related genes, particularly those involved in β-1,3-glucan and chitin synthesis, did not require functional MpkA. In the Fujioka et al. 2007 study, the authors reported decreased gene expression of agsB in strains with mutant mpkA. Our study did not find any significant changes in agsB expression through the time course investigated. These differences could be attributed to different genetic backgrounds, growth media, concentration of micafungin, and gene expression detection methods. Our results further underscore the complexity of the A. nidulans response, as we observed a significant transcriptional response to cell wall damage despite the absence of MpkA (~1000 genes compared to 1600 in wildtype). However, two features of the response in ΔmpkA mutants were notable. First, the timing of significant up-regulation of gene expression appears to be delayed by about ten minutes in ΔmpkA mutants relative to wildtype. Thus, a key feature of the MpkA-dependent component of the CWIS might be temporal, which would presumably ensure a rapid “all hands-on deck” response to repair the damaged cell wall. Second, certain classes of genes whose expression is up-regulated do in fact appear to be dependent upon the presence of functional MpkA. This includes sets of genes implicated in responses to light and in secondary metabolism. Other than the observation that their expression is induced by light, the function of most of the genes that fall into the former class remains unknown with the exception of brlA (see below) and conJ (AN5015) [25,26]. Amongst those genes that fall into the latter class, a small set of gene clusters potentially involved in secondary metabolite biosynthesis exhibited MpkA-dependent up-regulation in response to micafungin. On the other hand, another equally small set displayed MpkA-independent up-regulation. Other than underscoring the complexity of the transcriptional response to micafungin, the implications of the observed up-regulation of these gene clusters awaits their functional characterization. Of note, several additional gene clusters, including some encoding known polyketide synthases or non-ribosomal peptide synthases showed dynamic MpkA-dependent expression even though they were not induced by micafungin [27]. This presumably reflects the established role of MpkA in the regulation of secondary metabolism though the underlying mechanism remains to be determined [18,28].

Microcycle conidiation has been observed in a variety of filamentous fungi, and generally appears to serve as a “fail-safe” mechanism that enables fungi to efficiently divert resources from cellular growth to sporulation under adverse conditions [9,10]. The types of stress reported to trigger microcycle conidiation include nutrient depletion, high temperatures, altered pH, and high cell density [9]. The morphological steps that underlie microcycle conidiation vary across fungi, and relatively little is known about the precise mechanisms that subvert growth and cause sporulation. In A. nidulans, it has been previously shown that forced expression of brlA under conditions that normally do not promote sporulation results in the cessation of growth and the production of terminal conidia [15]. Although somewhat artificial, this form of microcycle conidiation requires the developmental regulators AbaA and WetA, as well as a functional Cdc42 GTPase module [15,16]. Other developmental regulators such as veA and wetA regulate microcycle conidiation in other fungi [11,29]. Here, we show that micafungin-induced expression of brlA coincides with the apparent occurrence of microcycle conidiation in A. nidulans. Although a functional test of the role played by BrlA in micafungin-induced microcycle conidiation awaits full investigation, preliminary analysis using the blrA42 mutant [30] revealed that it exhibits poor growth and no obvious morphological changes when exposed to micafungin (Appendix A). It is important to note here that our study did not detect significant changes in the expression of abaA or wetA during the 120 min period sampled following exposure to micafungin. However, we cannot exclude the possibility that expression did occur following this period but before morphological changes were observed at hyphal tips (i.e., starting at 180 min post micafungin exposure). Alternatively, it remains possible that the observed micafungin-induced morphological changes may not be dependent on all known components of the known regulatory pathways that control asexual development. Our results highlight the potential importance of microcycle conidiation as a survival strategy for fungi that encounter adverse environmental conditions. We propose that the CWIS is a multi-faceted response that is primarily dedicated to maintaining the integrity of growing hyphae through the repair of damaged cell walls. The existence of additional outputs such as microcycle conidiation conceivably provides alternative survival options should the damage be persistent or irreparable.

The echinocandins are an efficacious class of anti-fungal drugs that include micafungin [31]. However, the usefulness of the echinocandins is threatened by the emergence of resistance in fungi such as Candida species, and the genetic analysis of resistant isolates shows that they often harbor mutations in genes other than the known target FKS1 [32,33]. These other genes appear to primarily control processes that compensate form reduced cell wall β-glucans. Our results imply that microcycle conidiation represents another compensatory mechanism that might contribute to echinocandin resistance and suggest that further mechanistic study of this survival strategy might facilitate discovery of new approaches for countering resistance.

## Figures and Tables

**Figure 1 jof-07-00525-f001:**
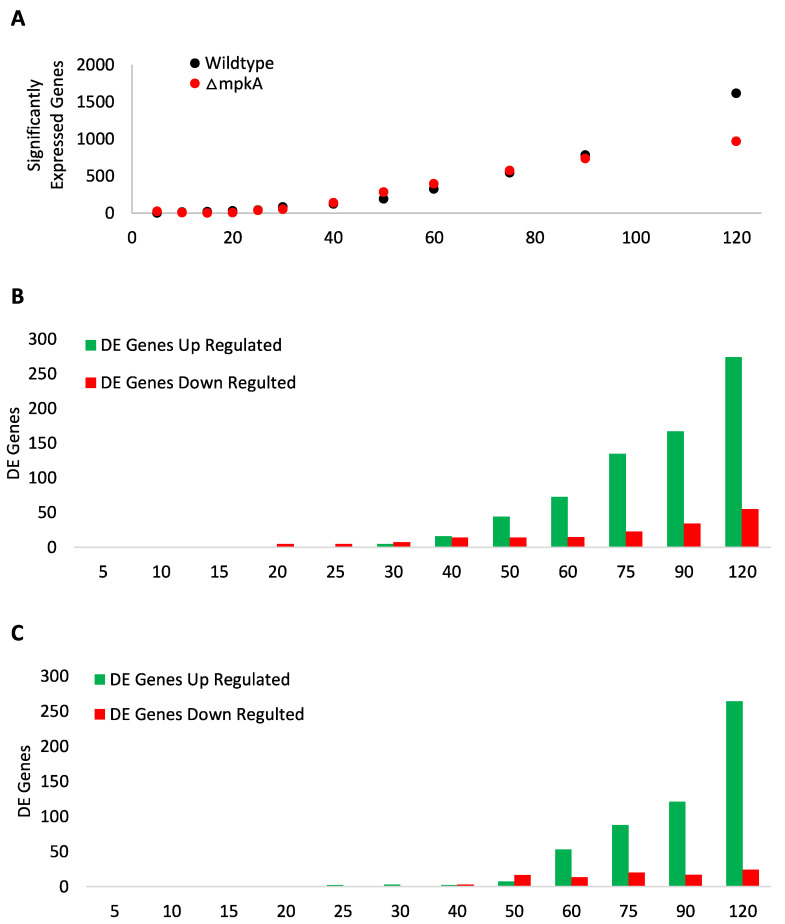
Global Gene Expression (**A**) Wild type showed exponential increase in significantly expressed genes post micafungin exposure. ΔmpkA showed linear increase in significantly expressed genes post micafungin exposure. Data points for both up-regulated and down-regulated genes. (Significantly expressed genes = *p*-value < 0.1). Three biological replicates. (**B**) Wildtype DE genes. The first DE gene in wild type was at the 15 min. (**C**) ΔmpkA DE genes. The first DE gene in ΔmpkA was at 25 min. (DE genes = adjusted *p*-value < 0.1, Log2Fold change (+/−) 2.0) Three biological replicates. *X*-axis is minutes.

**Figure 2 jof-07-00525-f002:**
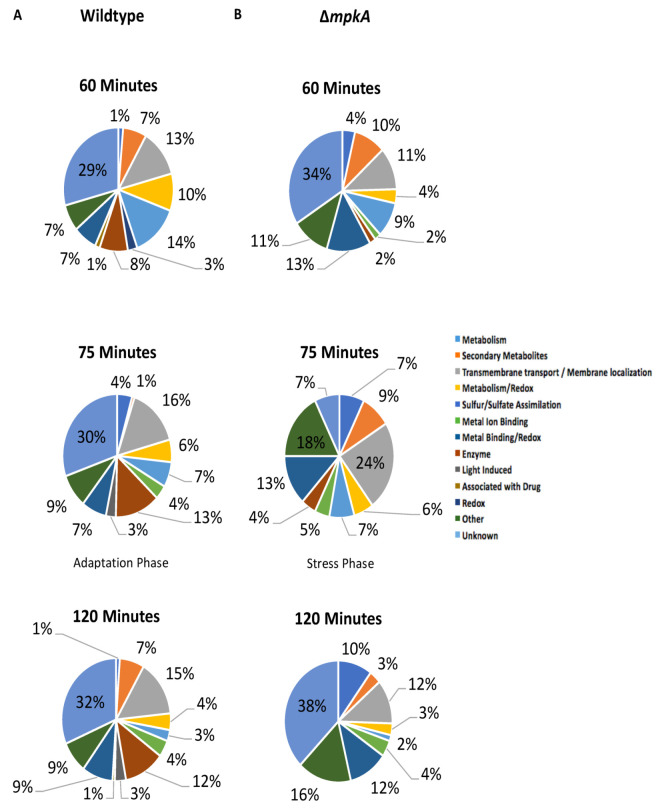
Gene Expression Shifts at the 75 Minute Time Point in Wildtype (**A**) Up-regulated wildtype gene expression at time points 60, 75, and 90-min (**B**) Up-regulated ΔmpkA gene expression at time points 60, 75, and 90-min. GO Terms from *Aspergillus* Database.

**Figure 3 jof-07-00525-f003:**
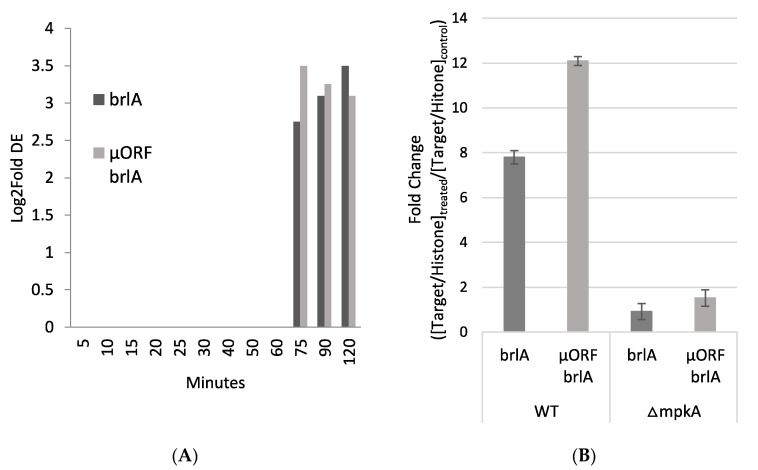
*brlA* Up-regulated in Wild Type but not ΔmpkA (**A**) RNA-Sequencing data showing *brlA* and *brlA* μORF up-regulated in wild type starting the adaptation phase. Three biological replicates. Significance DE = Log2Fold > 2.0. (**B**) qRT-PCR at the last time point, *brlA* and *brlA* μORF are up-regulated in wild type but not in ΔmpkA. Three biological replicates.

**Figure 4 jof-07-00525-f004:**
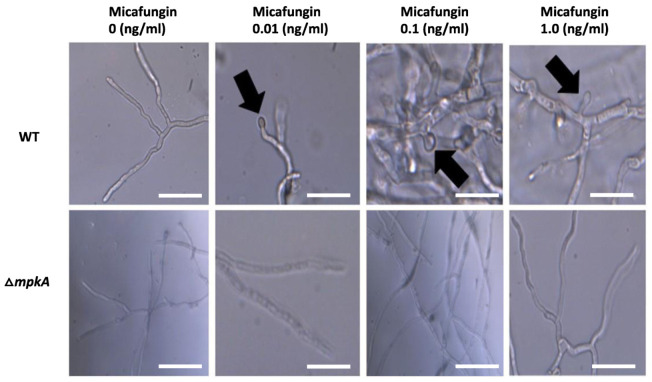
Hyphal morphology following exposure to micafungin. The morphology of wildtype and ΔmpkA hyphae that were untreated was normal. Wildtype hyphae display bulges at hyphal tips and lateral branches when exposed to micafungin at each tested dosage (black arrows). No morphological differences in ΔmpkA hyphae following micafungin exposure. 60× magnification. Scale bar is 25 µm. Pictures taken on Olympus BX51 microscope.

**Figure 5 jof-07-00525-f005:**
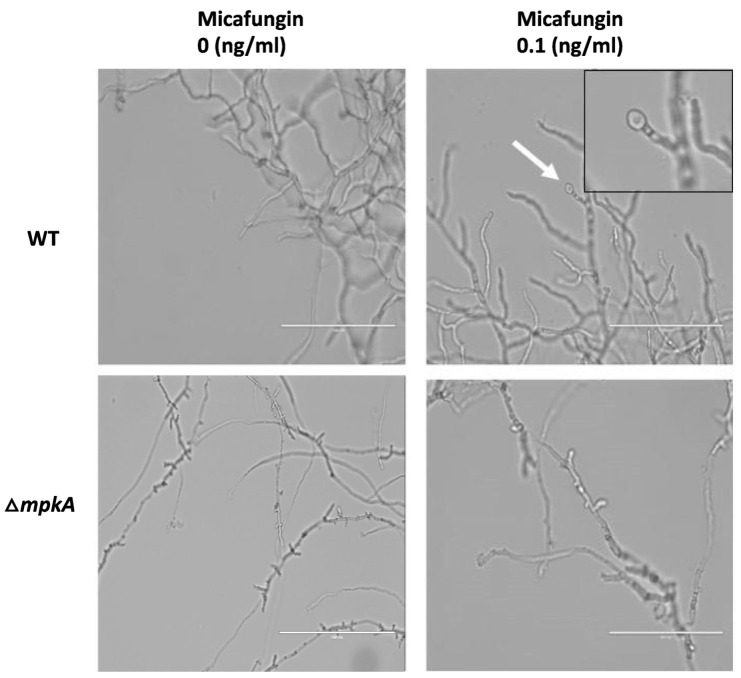
Development of terminal hyphal bulges following exposure to micafungin. Terminal bulges that resemble spores develop at hyphal tips in wildtype 9 h following exposure to micafungin. No bulges were observed in the ΔmpkA strain. Scale bar is 100 μm. Images captured on an EVOS FL microscope.

**Figure 6 jof-07-00525-f006:**
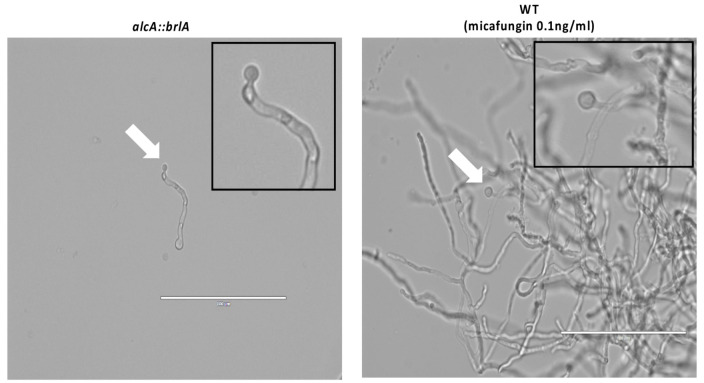
Similar Microcycle Spore Generation Similar spore phenotypes in up-regulated *brlA* strain after three hours promoter induction and wildtype nine hours post micafungin exposure (white arrows). 40× magnification. Scale bar is 100 µm. Pictures taken on EVOS FL microscope.

## Data Availability

RNA-Sequencing data deposited to NCBI and BioProject numbers are: PRJNA562694 and PRJNA727274. Additional information needed contact the corresponding author.

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
