# Peer review of "Micafungin-Induced Cell Wall Damage Stimulates Morphological Changes Consistent with Microcycle Conidiation in Aspergillus nidulans"

_jof, 2021, doi:10.3390/jof7070525_

Round 1

Reviewer 1 Report

This manuscript addresses the microcycle conidiation caused by micafungin, an inhibiter of cell wall beta-1,3-glucan synthesis, in Aspergillus nidulans. Initially, authors investigated the transcriptional response to the micafungin in A. nidulans wildtype and ΔmpkA strains by RNA-seq and found that the mpkA disruption abolished the transcriptional induction of the master regulator of conidia formation, brlA, and also the microcycle conidiation in the ΔmpkA strain. Authors confirmed the relationships between brlA and the microcycle conidiation by microscopic observation of the wildtype and alcA::brlA (a brlA conditional expression strain). The method is appropriate and the result is well discussed. In addition, I think the time course RNA-seq analysis should be highly evaluated.

I describe several comments as below.

  • Figures 4 and 5 and 6: How many bulges and/or conidia were observed per hyphal tips? It would be better to show the results quantitatively.

  • The alpha-glucan synthase encoding agsB shows deficient transcriptional induction in A. nidulans ΔmpkA strain treated by micafungin (Fujioka et al. 2007 [reference no. 3]). Please confirm whether authors observed the similar response or not, and describe it.

  • How about measuring the survival rate of the wild type, ΔmpkA, alcA::brlA strains after the treatment with or without micafungin. It will directly link between the survivability and the microcycle conidiation? Please perform the additional experiment if possible.

  • L140: Why would the authors choose p<0.1 as the criteria for significant change? Is this criterion a commonly used standard for RNA-seq? I feel we usually use p<0.05.

  • Please add a brief explanation about “brlA microORF” somewhere in text.

  • Abstract (L20-22): Although I’m not a native English speaker, I feel the sentence is something wrong. Please confirm.

  • L57: In think the “internal” is unclear in this manuscript. What does “internal” mean?

  • L48: I could not understand why authors cited the refence no. 4, a review about the resistance to azole fungicides here. I could not find any information related to beta-1,3- glucan in the reference no. 4.

  • Please check the sentence spacing in many places in the text (e.g. L20, L40).

  • L46: beta

  • L92-93: should provide the concentration of micafungin as weight per volume.

  • L121, L122: what does the unit “mN” mean?

  • I recommend to show the locus tags (e.g. AN7708, AN3043, AN9375 in L174) with putative functions. It would help authors to understand the significance of gene expression change.

  • Please correct the space between the number and the units in many places of the manuscript.

  • Figure 3 legend: “3 biological replicates” “Three biological replicates”

  • L215: italicize “brlA”

  • Reference: scientific names (e.g. Aspergillus nidulans) should be italicized.

Author Response

Dear Reviewer,

Thank you for taking the time to carefully review our manuscript and to provide helpful comments that greatly improved our paper. Please find the itemized responses to your comments. We hope these edits satisfy your concerns and have improved the manuscript to a point to where it is now suitable for publication.

  • Figures 4 and 5 and 6: How many bulges and/or conidia were observed per hyphal tips? It would be better to show the results quantitatively.

Note to reviewer: Added text in the results section about the frequency of microcycle conidiation spore generation “The formation of conidia via microcycle conidiation was relatively infrequent, with generally less than 5% of the hyphal tips differentiating into spores.” We believe the pictures in Figure 4, 5, and 6 are needed to represent the morphological changes in WT after exposure to micafungin. The addition of a graph representing the quantity of microcycle spore formation would be redundant now with the addition of the sentence explaining spore frequency added to the text.

  • The alpha-glucan synthase encoding agsB shows deficient transcriptional induction in A. nidulans ΔmpkA strain treated by micafungin (Fujioka et al. 2007 [reference no. 3]). Please confirm whether authors observed the similar response or not, and describe it.

Note to reviewer: Added text explaining finding differences in Fujioka et at. 2007 and our study in the discussion section “In the Fujioka et al. 2007 study, the authors reported decreased gene expression of agsB in strains with mutant mpkA. Our study did not find any significant changes in agsB expression through the time course investigated. These differences could be attributed to different genetic backgrounds, growth media, concentration of micafungin, and gene expression detection methods.”

  • How about measuring the survival rate of the wild type, ΔmpkA, alcA::brlA strains after the treatment with or without micafungin. It will directly link between the survivability and the microcycle conidiation? Please perform the additional experiment if possible.

 Note to reviewer: We agree with this point, and in fact have followed our observation with a comprehensive study of the viability of mpkA mutants, as well as other signaling mutants, in response to exposure to micafungin and other cell wall perturbing agents. A manuscript describing these new data is about to be submitted. We also note that the sensitivity of the mpkA deletion mutant was reported in Fig. 8 of Chelius et al., 2020. Lastly, it is currently not possible to test the effects of micafungin on the alcA::brlA strain, as the induction of alcA arrests growth of the strain and precludes a careful analysis of micafungin sensitivity.

  • L140: Why would the authors choose p<0.1 as the criteria for significant change? Is this criterion a commonly used standard for RNA-seq? I feel we usually use p<0.05.

Note to Reviewer: Sentence added in methods section explaining the logic behind choosing the adjusted p-value ≤ of 0.1 “Significance for the RNA-Sequencing gene expression was also based on our previous paper with a less strenuous adjusted p-value of ≤0. This was done to facilitate the incorporation of proteomic and transcriptomic data into a predictive gene expression model”.

  • Please add a brief explanation about “brlA microORF” somewhere in text.

Note to reviewer: Sentence added in text results section “brlA mORF is a small ORF located upstream ORF ofbrlA in a leader sequence. The mORF regulates conidial development by repressing the beta transcript of brlA when external cues trigger the initiation of asexual development (Han et al. 1993)”.

  • Abstract (L20-22): Although I’m not a native English speaker, I feel the sentence is something wrong. Please confirm.

 Note to Reviewer: Corrected in text

  • L57: In think the “internal” is unclear in this manuscript. What does “internal” mean?

 Note to Reviewer: The term “internal” was removed from text

  • L48: I could not understand why authors cited the refence no. 4, a review about the resistance to azole fungicides here. I could not find any information related to beta-1,3- glucan in the reference no. 4.

Note to Reviewer: A different reference was found and corrected in text “Walker, L.A., Gow, N.A. and Munro, C.A., 2010. Fungal echinocandin resistance. Fungal Genetics and Biology47(2), pp.117-126.”.

  • Please check the sentence spacing in many places in the text (e.g. L20, L40).

 Note to Reviewer: Corrected in text

  • L46: beta

 Note to reviewer: Corrected in text

  • L92-93: should provide the concentration of micafungin as weight per volume.

 Note to Reviewer: The concertation used was 0.1ng/ml and corrected in text.

  • L121, L122: what does the unit “mN” mean?

 Note to Reviewer: Concentration corrected to “mM” in text.

  • I recommend to show the locus tags (e.g. AN7708, AN3043, AN9375 in L174) with putative functions. It would help authors to understand the significance of gene expression change.

Note to reviewer: Gene names were added to ANIDs that had locus tags (ie changed AN5015 to conJ), and punitive function was added after ANID/gene name in the results section “From the 25-minute time point to the 60-minute time point, the data revealed a response to micafungin that featured increases in oxidoreductase activity genes including; AN7708 (predicted NADP+ 1-oxidoreductase activity) (Log2Fold 2.25), AN3043 (predicted role in oxidation-reduction process) (Log2Fold 2.66), AN9375 (predicted oxidoreductase activity and role in oxidation-reduction process) (Log2Fold 2.71), uaZ (urate oxidase) (Log2Fold 2.19), and aoxA1 (urate oxidase) (Log2Fold 2.90) (Figure 2A and Supplementary Table 8). However, at the 75-minute time point, additional GO term categories such as light induced genes (brlA ORF (brlA regulator leader sequence), brlA(zinc finger transcription factor), prtA (thermostable alkaline protease), and AN0045 (uncharacterized)) are up-regulated. The last time point had nine light-induced DE up-regulated genes (brlA, prtA, AN0045, AN0693 (uncharacterized), silA (uncharacterized), AN3304 (GABA transporter), AN3872 (uncharacterized), conJ (conidiation gene), AN8641(uncharacterized) (Figure 2A and Supplementary Table 8). Also, there is a known connection with secondary metabolism and light induced genes, which is particularly evident at the 75-minute time point and beyond. Examples mdpE (zinc finger transcription) (Log2Fold 2.55), pkdA (polyketide synthase) (Log2Fold 4.31), AN6962 (predicted secondary metabolism gene cluster member) (Log2Fold 4.38), AN9314 (protein with homology to entkaurene synthases) (Log2Fold 5.79), and AN2116 (predicted catalytic activity) (Log2Fold 6.34) (Figure 2A and Supplementary Table 8) [21]. Notably, the distribution of DE up-regulated GO terms for the ΔmpkA mutant does not display this apparent shift, as there are no light-induced up-regulated in any of the time points of the ΔmpkA strain (Figure 2A and Supplementary Table 8).”

  • Please correct the space between the number and the units in many places of the manuscript.

 Note to reviewer: corrected in text

  • Figure 3 legend: “3 biological replicates” “Three biological replicates”

 Note to reviewer: corrected in text

  • L215: italicize “brlA”

 Note to reviewer: corrected in text

  • Reference: scientific names (e.g. Aspergillus nidulans) should be italicized.

Note to reviewer: Corrected in text

Reviewer 2 Report

The work describes an investigation of the effects of micafungin treatment on the model fungus Aspergillus nidulans. The stated objectives are to examine cell wall induced signalling and understand the process of microcycle conidiation, both of which are not well understood. The approach taken is to use expression profiling (RNAseq) to document the early changes in gene expression upon treatment with micafungin. The data generated is generally quite interesting although not extensively characterised in the current manuscript. It was noted that micafungin treatment led to hyphal bulges that resembled microconidia and this was investigated in more detail to establish if this was indeed so. The presented evidence is consistent but circumstantial. A number of comments are provided below for the authors to consider.

1. On line 140, and elsewhere, it is stated that a p-value≤0.1 was used for differentially expressed genes. Is this correct? This is not a very stringent cut-off and the norm is usually at least 0.05. Given this, how many of the DEGs were validated by qPCR? What would the results look like with a p≤0.05? Was an FDR calculated? 

2. On lines 153-168 gene clusters are defined as three or more consecutive DEGs. Are these genes of related function or simply co-located? Is synteny of these conserved in closely related species (potentially pointing to a conserved mechanism)? Are the number of such DEG cluster statistically significant? The fact that cluster number increases with increased micafungin exposure, as does the actual number of DEGs, might suggest it is just random. Some analysis and support for this observation would be useful.

3. In section 3.3 are listed genes that are said to be examples of genes involved in secondary metabolism include pkdA, catA, amyF, AN9314, and AN2116. While AN9314 is clearly a gene that fits this description, the other genes are not. Maybe this list doesn't relate to the previous sentence that implies secondary metabolic function? If so, this section needs clarifying.

4. The suggestion that micafungin treatment induces microcycle conidiation is circumstantial. While brlA expression is up under these condition it is stated that abaA and wetA are not. There is no evidence that conidia can be formed in A. nidulans in the absence of abaA activity and ectopic over-expression of brlA in an abaA mutant strain supports this notion (Adams et al, 1988). The images provided in Fig 5 and 6 are not clear enough to distinguish between swollen tips of hyphae and actual conidia and the comparison between alcA::brlA and micafungin treatment in Fig 6 suggests the structures are different. If this tip morphology is truly indicative of conidiation, then testing the effects of micafungin in a brlA mutant would be a sensible test of the hypothesis. 

Author Response

Dear Reviewer,

Thank you for taking the time to carefully review our manuscript and to provide helpful comments that greatly improved our paper. Please find the itemized responses to your comments. We hope these edits satisfy your concerns and have improved the manuscript to a point to where it is now suitable for publication.

  1. On line 140, and elsewhere, it is stated that a p-value≤0.1 was used for differentially expressed genes. Is this correct? This is not a very stringent cut-off and the norm is usually at least 0.05. Given this, how many of the DEGs were validated by qPCR? What would the results look like with a p≤0.05? Was an FDR calculated? 

Note to reviewer: Sentence added in methods section explaining the logic behind choosing the adjusted p-value ≤ of 0.1 “Significance for the RNA-Sequencing gene expression was also based on our previous paper with a less strenuous adjusted p-value of ≤0.1. This was done to facilitate the incorporation of proteomic and transcriptomic data into a predictive gene expression model”.

The following sentences were also added to address the FDR calculation; “DESeq2 1.20.1 was the bioinformatics package used to analyze the read counts generated by HISAT2 2.1.0 and HTSeq-Counts 0.9.1. The FDR is calculated for every gene though DESeq2 at every time point and is called “adjusted p-value”. The adjusted p-value is what was used in addition of the Log2Fold(2 +/-) cut off to generated the DE genes list used in this study’s investigation.”

Logic behind choosing adjusted p-value ≤ of 0.1: To generate enough data points using both proteomics and transcriptomic analysis (Chelius et al. 2020 and this study) for our mathematic model that is in development, we needed to reduce the stringency from both studies from adjusted p-value of ≤0.5 to adjusted p-value of ≤0.1. In our Chelius et al. 2020 paper, multiple significant DE genes at 0, 60, 90, and 120 minute time points were verified through qRT-PCR.

  1. On lines 153-168 gene clusters are defined as three or more consecutive DEGs. Are these genes of related function or simply co-located? Is synteny of these conserved in closely related species (potentially pointing to a conserved mechanism)? Are the number of such DEG cluster statistically significant? The fact that cluster number increases with increased micafungin exposure, as does the actual number of DEGs, might suggest it is just random. Some analysis and support for this observation would be useful.

The gene clusters were identified on the basis of co-location (as noted in the text, “defined as three or more genes with consecutive ANID numbers”). We did not investigate the degree to which the clusters display synteny, though we recognize that this would provide additional insight into their putative functions. In addition, we recognize that the number of gene clusters identified will of course increase as the number of DE genes also goes up. The primary take-away from this analysis is qualitative -- it does no more than further emphasize the MpkA-dependent and -independent features of the CWIS. In particular, of the seven clusters whose expression is induced by micafungin, three are MpkA-dependent whereas four are not. Moreover, five of these seven genes respond relatively early (i.e., within the first 60-75 minutes post-exposure). This sentence has been added to the text.

The section has been revised as follows;

3.2 Gene Cluster Regulation

            Analysis of DE genes in both wildtype and the ΔmpkA strain at each time point revealed that a number of apparent gene clusters (defined as three or more genes with consecutive ANID numbers) were differentially regulated upon exposure to micafungin (Supplementary Table 7). For example, as early as 40-minutes post micafungin exposure, expression of the AN5269-5273 cluster was up-regulated in an MpkA-dependent manner whereas that of the AN7952-7955 cluster was up-regulated even in the absence of MpkA. In total, ten of the 19 presumptive gene clusters could be sorted into three groups whose expression was affected by micafungin; (i) MpkA-dependent micafungin induced (three), (ii) MpkA-independent micafungin-induced (four), and (iii) MpkA-dependent micafungin-repressed (three). Moreover, seven of these ten DE clusters responded within the first 60-75 minutes post-exposure. The remaining nine DE clusters were dependent on MpkA for expression but were not affected by exposure to micafungin, and were similar to those identified in our earlier study of MpkA-dependent gene expression in the absence of cell wall stress [18]. With some exceptions (Andersen et al., 2013), the precise function of the apparent micafungin-induced gene clusters is unknown, but this observation highlights the complexity of the CWIS and the presence of both MpkA-dependent and -independent components [10].

  1. In section 3.3 are listed genes that are said to be examples of genes involved in secondary metabolism include pkdA, catA, amyF, AN9314, and AN2116. While AN9314 is clearly a gene that fits this description, the other genes are not. Maybe this list doesn't relate to the previous sentence that implies secondary metabolic function? If so, this section needs clarifying.

Note to reviewer: The removal of catA and amyF with the addition of mdpE and AN6962 better define the types of secondary metabolism genes up regulated in WT after micafungin exposure. The gene GO terms were defined by Aspergillus Database. Text was changed to “Also, there is a known connection with secondary metabolism and light induced genes, which is particularly evident at the 75-minute time point and beyond. Examples include mdpE (zinc finger transcription) (Log2Fold 2.55), pkdA (polyketide synthase) (Log2Fold 4.31), AN6962 (predicted secondary metabolism gene cluster member) (Log2Fold 4.38), AN9314 (protein with homology to entkaurene synthases) (Log2Fold 5.79), and AN2116 (predicted catalytic activity) (Log2Fold 6.34) (Figure 2A and Supplementary Table 8).”

  1. The suggestion that micafungin treatment induces microcycle conidiation is circumstantial. While brlA expression is up under these condition it is stated that abaA and wetA are not. There is no evidence that conidia can be formed in A. nidulans in the absence of abaA activity and ectopic over-expression of brlA in an abaA mutant strain supports this notion (Adams et al, 1988). The images provided in Fig 5 and 6 are not clear enough to distinguish between swollen tips of hyphae and actual conidia and the comparison between alcA::brlA and micafungin treatment in Fig 6 suggests the structures are different. If this tip morphology is truly indicative of conidiation, then testing the effects of micafungin in a brlA mutant would be a sensible test of the hypothesis. 

Note to reviewer: Sentence added in discussion section explaining abaA and wetA gene regulation in our study “It is important to note here the expression of abaA and wetA is also involved in proper asexual development in A. nidulans (Adams et al., 1988). Our study did not detect significant changes in the expression of either gene during the 120-minute period sampled following exposure to micafungin. However, we cannot exclude the possibility that expression did occur following this period but before morphological changes were observed at hyphal tips (i.e., starting at 180 minutes post micafungin exposure. Alternatively, it remains possible that micafungin-induced microcycle conidiation may not be dependent on all known components of the known regulatory pathways that control asexual development.”

Different images were selected for Figures 5 and 6 to better represent the morphology of the spores in WT after micafungin exposure with the removal of the calcofluor experiments because the fluorescent images were too dark.

Text was added to the discussion section explaining preliminary data from brlA mutant experiments “A preliminary study using the blrA42 mutant revealed that it exhibits poor growth and showed no obvious morphological changes when exposed to micafungin (data not shown) (Gems et al. 1994).”

Round 2

Reviewer 2 Report

The revised manuscript has addressed most of the issued raised in the review of the initial submission. The following comments pertain to this revision.

1. The description of co-regulated clustered genes is fine but I have concerns about the use of the term 'clustered genes'. This term implies related function of the co-regulated genes in one location and is used quite specifically in the literature.

2. The issue of microconidiation remains unresolved. The new images clearly confirm what was evident in the original images that the tip bulges do not have the characteristic neck that is evident at the distal tip of phialides on conidiophores or on conidiating hyphae when brlA is ectopically expressed. Given that microconidiation features in the title and abstract as a major finding of this work, the burden of proof has not been met by the current dataset. The brlA42 temperature sensitive allele is leaky and still makes conidiophores at the restrictive temperature, so it might be expected to also make 'microconidia' under micafungin conditions. The fact that it is stated to have a growth defect under these conditions makes the interpretation of the result difficult (although interesting), as does the fact that the data are not presented. In the absence of confirmatory data the heavy emphasis of this phenotype in the title and throughout the manuscript is precarious.
